# Recommendations for AI in Gulf Digital Humanities

**Introduction**

In recent years, artificial intelligence—particularly generative AI—has increasingly drawn public attention, eliciting both enthusiasm and apprehension across various sectors. Within academic and cultural institutions, these emerging technologies have sparked debate about their impact on core scholarly missions, such as the preservation of knowledge, the promotion of critical inquiry, and the assurance of authenticity in cultural interpretation. In the nations of the Arabian Gulf region , these discussions take on heightened importance due to the region's unique position at the crossroads of technological ambition and deep-rooted cultural complexity. This paper argues that to promote inclusivity, accuracy, and cultural sovereignty, artificial intelligence applications in the Arabian Gulf's cultural heritage and digital humanities sectors must be ethically implemented, locally grounded, and co-developed within and across national communities.

Indeed, the Gulf, with more than 25 million digitized pages, 2.2 billion words of pre-modern Arabic texts in computational corpora, and projects spanning 3D documentation, AI-enhanced OCR[1], spatial humanities, and multilingual NLP, has launched an array of digital humanities initiatives across institutions in the whole region[2]. These efforts reflect existing commitment from dozens of institutions, including national archives, universities, and museums, to preserve and reinterpret heritage across at least ten languages and multiple religious, ethnic, and oral traditions. Strategic state-driven visions like Saudi Arabia's Vision 2030 and the UAE's National AI Strategy[3] have further propelled investment into digital infrastructure, making the Gulf not only a fast-moving site of innovation but a norm-setter for new standards in digital heritage.

At the same time, profound challenges persist: Western-centric training datasets, fragmented regulatory approaches, and entrenched epistemologies of ignorance within AI models all risk distorting local histories and marginalizing diverse voices. As Gulf cultural and academic institutions engage with generative technologies to classify, synthesize, or reconstruct heritage materials, they also navigate not only technical hurdles, but also deeper questions of governance, provenance, consent, cultural sovereignty, and meaning-making[4].

This paper responds by proposing a framework for community-driven, contextually grounded, and legally attuned AI development—one that treats Gulf heritage not as inert data to be extracted from the region, but as a living, sometimes contested, and evolving foundation for reimagining digital humanities.

**The Gulf Context: Opportunity and Responsibility**

The Arabian Gulf, with its dynamic cultural landscape, deep-rooted oral traditions, and growing digital infrastructure, offers both a rich repository of materials and a critical testing ground for new technologies. Over the past decade, there has been a noticeable shift from traditional heritage practices to the adoption of computational approaches, giving rise to what may now be termed "Computational Gulf Studies". This interdisciplinary field, in dialogue with global conversations of the computational humanities, blends digital humanities employing AI methods such as handwritten text recognition (HTR), natural language processing (NLP), computer vision (CV) tools and multimodal models like

---

[1] B. Kiessling, K. Gennady, M. Miller, and K. Smail. 2021. ""Advances and Limitations in Open Source Arabic-Script OCR: A Case Study," Digital Studies/Le champ numérique 11(1): 8, pp. 1–30; A. Heakl et al., "KITAB-Bench: A Comprehensive Multi-Domain Benchmark for Arabic OCR and Document Understanding," arXiv preprint (2024). There have also been community based models trained specifically on Gulf historical handwritten documents as demonstrated by "Khat," a Arabic Transkribus model (2024).

[2] Examples of large-scale initiatives include: the OpenITI Corpus, which contains over 2 billion words of pre-modern Arabic texts; the Qatar Digital Library, hosting more than 2 million digitized pages of Gulf and Islamic history; KAUST's archaeological AI research using satellite imagery to detect ancient stone structures; the Digital Sirah Project by Qatar National Library and Aga Khan University; and NYU Abu Dhabi's Dhakira Center, Music and Sound Cultures group, and OpenGulf research group , all of which explore AI and digital tools for heritage analysis and preservation. Projects like ArtELingo at KAUST also demonstrate efforts in multilingual NLP and emotional AI tailored to Arabic cultural contexts.

[3] See: Ministry of Culture, Saudi Arabia, Our Cultural Vision; Also: UAE National AI Strategy.

[4] N. Catelan, X. Fresquet, S. Moura, P. Svärd, D. J. Wrisley. AI and Digital Humanities in the Arabian Gulf: Interdisciplinary Perspectives on Infrastructure, Cultural Heritage, and Community Building. Intelligence artificielle et humanités numériques dans le Golfe arabo-persique : perspectives interdisciplinaires sur l'éthique, le patrimoine culturel et la construction communautaire, Sorbonne Abu Dhabi, Feb 2025, Abu Dhabi, France. ⟨hal-05065443⟩

CLIP (Contrastive Language–Image Pretraining) with conversations about critical AI[5]. These mixed methods approaches are increasingly applied to vast collections of archival documents, audiovisual materials, and museum holdings, offering opportunities for research and public engagement, even redefining the notion of a collection itself.

As an example, the Sharjah Metaverse Platform is experimenting with immersive digital reconstructions of architectural and intangible heritage[6], while the Qasr Al Hosn archive in Abu Dhabi has begun incorporating NLP techniques to enrich metadata attached to historical documents. These initiatives signify a decisive shift from analog heritage stewardship to AI-driven innovation.

However, these innovations are not without risk. Models not attuned to local specificities often perpetuate biases, which often carry implicit classification, and representation. When deployed without local adaptation, these models can produce misleading or culturally insensitive outputs. Examples of such undesirable outputs abound. A Nabati poem, part of a Bedouin oral tradition in the Emirates, which was misclassified by a CLIP model as an "Orientalist chant", visual misclassified Ramadan decorations among Christmas ornaments, images of Emirati men engaged in falconry in a desert landscape were miscaptioned as a family sitting on a beach and similarly, traditional Emirati jewelry digitized by museums in Qatar was auto-labeled by image classifiers as "tribal costume accessories," a term loaded with colonial undertones[7]. The stakes of such computational judgment can be higher in textual content riddled with OCR error, where transcription errors obscure nuanced distinctions, highlighting the need for critical oversight.[8]

These misclassifications are not outliers and they are also not trivial—they shape how cultural artifacts are presented, interpreted, and valued by both local communities and international audiences. They also reinforce historical asymmetries in global knowledge production by positioning Gulf heritage through a distorted lens, and in doing so, perhaps more importantly, they can erode public confidence in the benefits of AI in society at large. They call for increased local participation in model training. A recent project at Zayed University revealed how generative image models repeatedly defaulted to desert imagery and opulent skylines when asked to depict Emirati culture—prompting the development of a new visual database, *The Dis-Orientalist*, challenging stereotypical AI-generated representations[9].

## Counter-Archives: Ethics, Resistance, and AI in Gulf Heritage Studies

Traditional archival practices have historically privileged written, institutional, and male-dominated forms of documentation, systematically excluding others. In particular, intangible, oral, and gendered forms of knowledge—such as poetry, storytelling, music, craft traditions, and embodied memory—have often been deemed unofficial, ephemeral, or less legitimate than textual records. This exclusion is especially significant in regions like the Gulf, where oral transmission and collective memory play a central role in cultural identity[10]. As a result, entire segments of historical experience—particularly those of women, minority communities, and non-elite actors—have remained underrepresented in mainstream archival collections and AI-augmented datasets.

Frameworks like feminist standpoint appraisal and postcolonial archival theory offer practical tools for rethinking heritage AI[11]. These approaches reject the idea of archival neutrality, showing how curatorial choices reflect social hierarchies and exclusions, and expose how colonial legacies continue to shape dominant archival norms. Both call for community-centered and plural epistemologies. These insights are not only abstract claims: they must inform how AI systems are built, whose histories they privilege, and how they navigate questions of representation and authority.

[5] L. Klein et al., "Provocations from the Humanities for Generative AI Research," arXiv:2502.19190v1 (2025); L. Tilton et al., eds. *Computational Humanities*. Minneapolis: University of Minnesota Press, 2024.
[6] American University of Sharjah, "AUS Actively Promotes Preservation of UAE's Cultural Heritage".
[7] Observer Research Foundation, "The Role of Digitalisation in Heritage Conservation".
[8] S. Kirmizialtin and D.J. Wrisley. "Exploring Gulf Manumission Documents with Word Vectors" *Journal of Digital Islamicate Research* 2.1-2(2024): 1-29.
[9] R. Bedirian, "'What is Abu Dhabiness?' Exhibition Challenges AI's Stereotypical View of the UAE at Venice Biennale." The National, May 11, 2025.
[10] K. Landström. Archives, Epistemic Injustice and Knowing the Past. Ethics and Social Welfare, 15.4(2021): 379-394.
[11] M. Caswell, Urgent Archives: Enacting Liberatory Memory Work (Routledge, 2021); J. Bastian, "Whispers in the Archives: Finding the Voices of the Colonized in the Records of the Colonizer," in Political Pressure and the Archival Record, ed. M. Procter, M. Cook, and C. Williams (Chicago: Society of American Archivists, 2006).

For example, a community-led initiative in Bahrain focused on the digitization of women's pearl-diving songs[12] has shown how collaborative efforts can counteract erasure: through active participation of local women, linguists, and musicologists, these recordings were annotated and contextualized in ways that enabled AI tagging systems to accurately recognize melodic structure, thematic content, and cultural significance. Rather than being misclassified as generic "folk chants" or filtered through orientalist tropes, these cultural expressions were embedded within meaningful, community-authenticated metadata that respected their origin and emotional weight.

Conversely, a dataset used to train an AI model on Gulf family histories was compiled using a generalized corpus labeled as "Middle Eastern genealogical texts." As a result, the system misrepresented Emirati tribal lineages, incorrectly associating them with unrelated Levantine clans based on shared names or linguistic similarities across Arabic-language societies. This not only led to factual inaccuracies, but also threatened to distort local historical narratives and undermine tribal, national and regional identities that remain vital in both social and political spheres. These types of algorithmic misreadings betray variable quantities of training data across the larger "Middle East" and reveal deep misalignments between the assumptions embedded in training data and the sociocultural realities they aim to reflect.

To move forward, AI development in archival contexts must engage directly with alternative epistemologies—those shaped by gendered experience, oral culture, and local memory. Cultural heritage institutions such as archives in the Gulf are particularly well-positioned to lead in this effort, given the region's long standing emphasis on intergenerational knowledge transmission. By embedding debates in critical AI and archival ethics into AI workflows—from data collection to model validation—these institutions can foster a more equitable and culturally resonant approach to digital heritage. In doing so, they not only correct historical omissions but also ensure that the next generation of AI tools is built on principles of justice, plurality, and respect. Recent efforts at NYU Abu Dhabi's Music and Sound Cultures group and the Dhakira Center also reflect this ethic, exploring AI-driven music analysis and digital heritage research that foreground regional and transnational perspectives[13].

Additionally, private and community archives—such as family collections, community-run digital platforms, or social media accounts—abound in the Gulf region and represent culturally significant, but often precarious repositories. AI systems must be designed not only to work with institutional archives, but also to respect the decentralized, fragile, and sometimes contested nature of these cultural materials. Moreover, as scholars have noted in indigenous studies in Canada and Australia, archival digitization without careful attention to data sovereignty can result in the reappropriation of community knowledge[14].

Questions of who stewards digitized materials, how knowledge is accessed and interpreted, and under what terms it may be disseminated must be central—not peripheral—to heritage AI strategy in the Gulf. It is critical to promote accuracy and reliable cultural heritage, protect community control, foster intellectual freedom, and acknowledge cultural sensitivities to ensure equitable participation in the future of digital humanities.

**Mind the Gap: Regulating AI Before It Rewrites History**

While global approaches to AI regulation diverge, the Gulf—particularly the UAE—presents a distinctive model of strategic regulatory minimalism. In contrast to the European Union's assertive, risk-tiered AI Act, the UAE has opted for soft-law instruments such as charters and strategic roadmaps, favoring flexibility and economic signaling over enforceable constraints. This regulatory

---

[12] Gillespie, C., & Ahmad, S. Z., "Cultivating Cultural Memory: A Case Study of the Revitalization of Al Jazeera Al Hamra Heritage Village," Gulf Education and Social Policy Review 6, no. 1 (2025): 3–26.

[13] Computational Audio Analysis for Cultural Heritage Preservation, Council on Undergraduate Research. This initiative, led by NYU Abu Dhabi's Music and Sound Cultures research group, explores the use of digital signal processing and AI tools to analyze, archive, and interpret musical heritage from the Gulf, Levant, East Africa, and South India. By combining ethnomusicology with computational methods, the project exemplifies interdisciplinary approaches to the preservation of intangible cultural heritage. Dhakira Center for Heritage Studies in the UAE, NYU Abu Dhabi. The Dhakira Center engages in interdisciplinary heritage research, connecting digital humanities methodologies with local memory practices and policy discourse. While not exclusively focused on AI, the center's emphasis on digital tools and archival theory provides a robust foundation for future integration of machine learning in Gulf heritage studies.

[14] T. Kukutai, J. Taylor (2016). Data Sovereignty for Indigenous Peoples: Current Practice and Future Needs. In Indigenous Data Sovereignty: Toward an Agenda. ANU Press.; Marsh, D. E. (2023). Digital Knowledge Sharing: Perspectives on Use, Impacts, Risks, and Best Practices According to Native American and Indigenous Community-Based Researchers. Archival Science, 23(1), 81-115.

posture is less about inaction than about positioning: AI governance is conceived as a lever for investment and productivity, not as a framework to curtail emerging risks.

A telling example is the UAE's 2024 announcement of a national law on quantum cryptography—issued well in advance of the technology's practical maturity. In this context, legal acts often serve symbolic or economic functions, indicating readiness and ambition rather than introducing mechanisms of control. More recently, the creation of a Regulatory Intelligence Office powered by AI[15] underscores an inversion of traditional governance: AI is not being regulated, but is being enlisted to regulate. This "Abu Dhabi doctrine" suggests a paradigm wherein policy is designed to attract and foster innovation rather than to constrain it preemptively .

Yet for cultural heritage and digital humanities, this posture creates an unstable operating terrain. Institutions are encouraged to digitize and experiment with AI-driven reconstructions of texts, artifacts, and oral traditions, but they do so in the absence of sector-specific regulation. Most heritage-related AI deployments are subject only to general-purpose laws—such as localized versions of the GDPR or intellectual property norms—which offer little preventive capacity or risk assessment. Crucially, this legal landscape lacks clarity on foundational issues—even as regional strategies like the Riyadh Charter for AI Ethics[16] and Qatar's AI Ethics and Governance Framework[17] signal a growing emphasis on localized ethical models that align with Islamic values and cultural priorities.

These omissions are consequential. AI-generated heritage narratives—such as avatars of historical figures, AI-curated exhibits, or predictive reconstructions of tribal genealogies—carry high cultural sensitivity. Misclassification or factual distortion risks not just reputational damage but also the erosion of cultural sovereignty and community trust. A recent controversy involving the HBKU AI Characters project, where personas were digitally recreated without transparent sourcing or community validation, exemplifies the potential fallout: while no laws were violated, the absence of procedural safeguards ignited public concern. One can imagine any one of the misclassifications and mislabelings mentioned in the first section of this paper creating a similar public affront.

What emerges is a regulatory grey zone, where innovation is prioritized, but community safeguards are underdeveloped. A hybrid model is urgently needed—one that combines European-style enforceability with Gulf-style agility and contextual sensitivity. Such a framework would: (1) Provide legal clarity over data provenance, reuse, and consent; (2) Promote transparency in heritage-related AI design and deployment; (3) Recognize the sector's non-commercial stakes: identity, memory, and representation.

**Cultural Institutions as Ethical Anchors in the AI–Heritage Innovation Ecosystem**

Galleries, Libraries, Archives, and Museums (the so-called GLAM institutions) play a pivotal role in shaping the ethical and epistemological foundations of AI applications in the heritage sector. As custodians of memory, they bring expertise and community trust that are essential for ensuring AI tools reflect diverse values, but they must work in tandem with academic institutions that also shape public understanding and cultural narratives. For instance, collaborations between museums in Abu Dhabi and local universities[18] have enabled AI-driven metadata enrichment that respects both artistic interpretation and provenance. Ultimately, memory institutions hold a fundamental role in establishing the ethical and normative frameworks for cultural AI applications within the whole sector. Concurrently, partnerships with commercial enterprises can contribute to the scaling of digitization and interface design, provided these are guided by frameworks grounded in cultural ethics as opposed to exclusively commercial imperatives. For example, initiatives at KAUST, including its hackathon

[15]J. Werner, "UAE Launches World's First AI-Powered Regulatory Intelligence Ecosystem." BABL AI, April 16, 2025.
[16]Islamic World Educational, Scientific and Cultural Organization (ICESCO). "ICESCO Director-General: Riyadh Charter on AI Ethics a Moral Compass Anchored in Islamic World Values." ICESCO, May 6, 2025.
[17]Sharek. "Artificial Intelligence in Qatar—Principles and Guidelines for Ethical Use." Hukoomi – Qatar Government e-Participation Portal, May 16, 2024.
[18]D. J. Wrisley, E. Guéville, and N. A. Cappelletto, "Creating New Audiences for Digital Objects Through University-Museum Collaboration," Museums in the MENA Region Journal 3 (2022): 61–63. See also: Sorbonne University Abu Dhabi, Digital Archiving in the Arab World (DAAW) 2024 Booklet.

addressing AI bias[19], and ethics training at NYU Abu Dhabi and Sultan Qaboos University[20], have helped frame AI ethics not just as compliance but as cultural stewardship.

These dilemmas are real and pressing. In the region, AI—especially generative models—is seen both as a tool for preserving heritage and for centralizing cultural narratives under state control. Ethical AI research must navigate these tensions, as AI can reveal or conceal resistance and diversity.

**Alternative Views: Innovation Without Localization?**

In debates around AI in cultural heritage, some argue that the rapid pace of innovation favors generalized, scalable models over localized, culturally tailored ones. Industry highlights the cost-efficiency and speed of using large pre-trained models like Llama, GPT-4 or CLIP "as-is," citing high expenses tied to regional models such as Jais[21] in the UAE. For many under-resourced institutions, these generalist models remain the only practical option despite cultural limitations.

Another perspective prioritizes access over precision, arguing that imperfect outputs are preferable to no engagement at all. Even if CLIP misclassified heritage items—labeling pearl-diving tools as "fishing gear"—the initiative was hailed as a success for raising public awareness and accelerating digitization efforts[22]. Techno-optimists further argue that as training datasets expand, generalist models will naturally improve in cultural nuance through sheer scale.

These views raise valid concerns, but risk normalizing misrepresentation. Cultural heritage is more than data—it shapes identity and memory. Mislabeling a tribal genealogy is not trivial; it rewrites history. Localization must be embraced as a site of innovation—one that challenges homogenizing systems and restores epistemic pluralism. Scholars warn of techno-determinism[23] and call for tools that are critically engaged, not assumed neutral. The Gulf can lead by treating localization not as a constraint, but as an ethical and methodological frontier.

Additionally, it is important to challenge the assumption that "openness" alone resolves issues of accessibility and fairness. Indeed, as Klein et al. have argued, "openness is not an easy fix,"[24] that is, without transparency about data origins, cultural context, and power dynamics it risks reinforcing existing inequalities rather than democratizing AI.

# Ten Recommendations
# for Ethical and Community-Guided AI in Gulf Heritage

### #1. Establish a Gulf AI & DH Knowledge Cities Consortium

Rationale: Fragmentation across institutions has led to redundant projects and siloed expertise. A federated structure can coordinate innovation, training and resource allocation while respecting institutional diversity.

Implementation: Support the emergence of interconnected "AI poles" in cities like Abu Dhabi, Dubai, Doha, Jeddah, and Muscat—each anchoring local expertise in DH and AI. This consortium would be coordinated via shared data standards, multilingual API infrastructure, and periodic convenings. Drawing on DARIAH's governance model, it would promote collaboration while allowing for differentiated local priorities.

Risk: Requires strong political and institutional buy-in; disparities in technical capacity and narrative orientation may complicate cross-pole coordination.

---

[19]Hackathon Tackles AI Bias with Inclusive Solutions, KAUST Innovation. This event brought together interdisciplinary teams to address algorithmic bias and develop inclusive AI tools tailored to the socio-cultural contexts of Saudi Arabia. By emphasizing fairness, representation, and transparency in model design, the hackathon demonstrated KAUST's commitment to fostering responsible AI development aligned with both global standards and regional ethical priorities.

[20]UNESCO Chair on Artificial Intelligence and Ethics, Sultan Qaboos University. This Chair focuses on advancing ethical research, policy engagement, and educational practices in AI across sectors such as education, medicine, and industry. It supports the implementation of UNESCO's global recommendations on the ethics of AI, with a specific emphasis on cultural and regional relevance in the Arab world.

[21]Sengupta, Neha, et al. "Jais and Jais-chat: Arabic-Centric Foundation and Instruction-Tuned Open Generative Large Language Models." arXiv preprint arXiv:2308.16149 [cs.CL], last revised September 29, 2023. https://doi.org/10.48550/arXiv.2308.16149.

[22] S. Thompson, "The Jazeera Al Hamra Digital Heritage Project," The International Journal of the Inclusive Museum 8, no. 3 (2015): 43–56.

[23] I. A. Bhat, S. A. Ganaie, "Digital Humanities in Cultural Preservation," in Handbook of Research on Inventive Digital Tools for Collection Management and Development in Modern Libraries (IGI Global, 2015), 323–340.

[24] L. Klein et al., "Provocations from the Humanities for Generative AI Research," arXiv:2502.19190v1 (2025).

## #2. Fund a Shared Regional DH+AI Infrastructure Hub

Rationale: A unified technical ecosystem can democratize access to tools and datasets, foster standardization, and reduce costs.

Implementation: Establish a federated infrastructure for LLMs, digitized archives, multimodal tools (e.g., eScriptorium, CLAP/CLIP), and regional models (e.g., Jais, Falcon, CAMeLBERT). This would include APIs, multilingual documentation, and cloud support via regional tech partners.

Risk: Requires high technical capacity and cybersecurity resilience; risks centralizing control over sensitive cultural data without adequate safeguards.

## #3. Develop Community-Validated Training Datasets

Rationale: Current datasets are often Western-centric, failing to capture Gulf-specific cultural nuances and oral traditions.

Implementation: Launch participatory dataset development programs involving poets, tribal historians, musicians, craftsmen, and oral archivists from across the region. These programs would be hosted at local universities and heritage centers, ensuring multi-generational input. Examples include the use of AI to digitize Omani folk stories for children at Sultan Qaboos University, and the multilingual datasets from KAUST that prioritize Arabic-language emotional modeling[25].

Risk: Community participation may be uneven across countries; the process can be time-intensive and costly, potentially excluding less-resourced communities or dialect groups.

## #4. Organize an Annual Gulf Conference on AI and DH

Rationale: No recurring, region-wide venue currently exists to consolidate discourse, training, and publication at the intersection of DH and AI.

Implementation: Establish an annual international conference rotating between GCC countries. Proceedings would be peer-reviewed, indexed, and published via open access. Thematic calls (e.g., AI and Maritime Heritage, NLP for Oral Traditions) would align with national strategies.

Risk: High organizational and financial burden may lead to uneven hosting capacity; conference discourse may be dominated by elite or political narratives.

## #5. AI talents in DH: Support Joint PhD and Postdoctoral Programs

Rationale: The region needs interdisciplinary experts who can bridge cultural studies, computer science, and policy.

Implementation: Fund co-supervised PhD and postdoctoral positions between universities[26] and cultural institutions[27]. Topics could range from AI-enhanced manuscript editing to ethical LLM applications in oral heritage.

Risk: Talent retention is a challenge; without clear career pathways, fellows may migrate abroad or exit the field after funding ends.

## #6. Create High School-Level Digital Humanities Literacy Programs

Rationale: Introducing digital humanities (DH) and AI literacy at the secondary school level builds foundational awareness of data ethics, computational tools, and cultural preservation, while fostering future talent across disciplines. Successful models of such integration exist in North America.Implementation: Ministries of Education should collaborate with universities, libraries, and cultural institutions to develop adaptable digital humanities and AI literacy modules for secondary schools. These would introduce students to ethical data use, AI narratives, and tools connecting Arabic, history, and computing. As a first step, annual workshops should train undergraduates and

---

[25]Preserving and Enhancing Heritage Through AI Technology, Oman Observer. The article highlights efforts at Sultan Qaboos University to digitize and visually reinterpret Omani folk stories for children. It underscores the role of AI in cultural transmission, education, and the preservation of intangible heritage within the context of Oman's national identity.

[26] In [pParticular with NYUAD, KAUST, MBZUAI, HBKU, SQU, AGU, SUAD, UAEU, KSU, AUS, and GU-Q.

[27] In particular with King Fahd National Library, National Museum of Qatar, Qatar National Library, King Abdulaziz Foundation for Research and Archives – Darah, Bahrain Authority for Culture and Antiquities, Louvre Abu Dhabi, King Abdulaziz Center for World Culture – Ithra (Dhahran), Museum of Islamic Art (Qatar), Sharjah Art Foundation, Sharjah Institute for Heritage, Jameel Arts Centre (Dubai), and the Royal Opera House Muscat Archives.

educators in AI basics, programming, prompt design, and classroom strategies. Events such as the NYU Abu Dhabi [Hackathon for Social Good in the Arab World](#) can serve as a model.

Risk: Curriculum adoption may vary across national systems and face political sensitivities around AI, ethics, or cultural heritage. Differences in digital infrastructure, teacher training, and language priorities may also affect implementation. To address this, pilot programs should be flexible and locally adaptable, enabling countries and institutions to tailor content while following common pedagogical principles.

## #7. Establish Ethical AI Charters for Cultural Heritage

Rationale: Most AI charters are overly general, often focused on data privacy, algorithmic fairness, or human oversight in commercial sectors. They rarely address the sector-specific challenges: provenance, consent, representational accuracy, narrative authority, or epistemic harm. In the Gulf context, where AI is being deployed for sensitive cultural content, an ethical vacuum increases the risk of misappropriation or distortion—especially in the absence of binding regulations.

Implementation: Develop a sector-specific Ethical AI Charter for Heritage and Archives, led by flagship institutions such as Darah (Saudi Arabia), the Qatar National Library, and the National Library and Archives of the UAE. The drafting process should involve: (1) Legal scholars with expertise in data sovereignty and cultural IP rights; (2) Technologists familiar with LLMs, image/text generation, and metadata systems; (3) Community representatives, especially from oral and performative traditions; (4) Ethicists and historians to ensure cultural resonance and narrative plurality. The charter should be modular and adaptive, offering practical guidelines on the use of AI in archival digitization, generative reconstructions, metadata tagging, and public-facing AI tools in museums. It should also recommend minimum disclosure standards for heritage-related AI models, regarding training data origins and cultural representation choices. The Riyadh Charter for AI Ethics offers a foundational regional precedent for such charters. Future charters should build upon this initiative while expanding focus to archival provenance, cultural IP, and heritage-specific disclosures.

Risk: The primary challenge is legal and institutional traction. Without formal regulatory backing or linkage to project funding criteria, the charter risks remaining symbolic. To increase enforceability, tie adherence to eligibility for public funding, international collaborations, or institutional accreditation. Moreover, ensure the charter is not static, but subject to periodic review by a transnational advisory board to reflect evolving technologies and cultural sensibilities. It should be (deemed as) a living and prescriptive document.

## #8. Promote Participatory AI Auditing in Heritage Projects

Rationale: In heritage contexts, AI evaluation must go beyond accuracy metrics to assess cultural appropriateness, representational fairness, and alignment with community values. Without this, systems risk perpetuating historical misrepresentations or introducing new forms of epistemic harm. Participatory auditing—where affected communities have a say in evaluating AI outputs—is essential to ensure legitimacy and trust.

Implementation: Integrate participatory review mechanisms into key stages of heritage-AI projects. Rather than large, formal panels, projects could implement: (1) Targeted community feedback sessions during pilot testing phases (e.g., with educators, artists, tribal historians, or youth groups); (2) Short-list audits where community members review a curated sample of AI outputs (e.g., generated descriptions, reconstructed visuals, tag suggestions); (3) Documentation of flagged issues and changes made as a result (to be included in project reports or exhibitions).

In larger national or institutional projects, more formal Community Advisory Boards can be constituted with clear mandates and time-limited scopes.

Risk: Participatory auditing may face logistical challenges, such as identifying representative and qualified community voices, compensating participants fairly, and handling disagreement. If not well-scoped, the process can become too procedural or spark resistance from institutions wary of external critique.

## #9. Embed Critical Appraisal in AI Design

Rationale: AI models often replicate existing social biases, including colonial hierarchies, gender marginalization, and urban or elite perspectives. These distortions result from systemic absences in the training data and unexamined assumptions during model development.

Implementation: Introduce a structured critical appraisal phase within the project cycle of heritage-related AI initiatives. It includes a standardized reflection document appended to each project proposal and final report. The document should address: (1) Whose perspectives are represented or absent in the dataset; (2) How decisions were made about what constitutes "relevant" data; (3) Whether oral, performative, or marginalized knowledge forms (e.g., women's stories, rural traditions, minority languages) were considered, and if not, why; (4) How misrepresentation risks were identified and mitigated.

Risk: Such processes may be perceived as burdensome, especially in fast-paced or technically oriented environments. Without institutional incentives (e.g., tying appraisal to grant eligibility or ethics approval), there is a risk the process becomes a symbolic exercise.

**#10. Support Multilingual and Multimodal AI Fine-Tuning**

Rationale: Gulf artifacts and traditions are frequently misclassified due to inadequate model localization.

Implementation: Use regional image, text, and audio collections to fine-tune models. Partner with GLAM institutions for training data and testing the models on samples of digitized collections.

Risk: High computational cost and legal/ethical constraints on sharing training data; model performance may still lag for underrepresented dialects and traditions.

**Conclusion**

The region of the Arabian Gulf presents a rare convergence of opportunity and responsibility. With world-leading investments in AI infrastructure, large multilingual and multimodal archives, and a deep reservoir of oral and performative traditions, the region is not merely adopting AI for cultural heritage—it is shaping how AI should ethically engage with the past. And indeed, Gulf institutions are already demonstrating that inclusive, community-guided, and culturally-situated approaches to AI are not only possible, but scalable.

However, leadership in this domain must be earned through sustained commitment to pluralism, regulation, and innovation. As this paper shows, off-the-shelf models often embed biases that lead to cultural misrepresentation. The Gulf's strategic choice is not whether to use AI, but how—and on whose terms. By embedding feminist and postcolonial ethics, investing in regional AI infrastructure, and designing participatory auditing mechanisms, the region can model a new standard: one that privileges cultural specificity over computational efficiency, and ethical co-authorship over extractive automation.

This vision is actionable. The ten recommendations proposed in this paper provide a coherent and pragmatic roadmap—ranging from the creation of regional coordination bodies to the development of digital humanities literacy programs. Yet they represent only an initial step. New and pressing questions demand deeper exploration: How can AI systems be designed to accommodate plural epistemologies while maintaining interoperability? What mechanisms can be developed to protect communal rights over digitized traditions in the absence of established legal frameworks? How might future Gulf-based large language models (LLMs) reflect not only linguistic fluency but also cultural sensitivity and narrative plurality? Can ethical charters be transformed into enforceable policies?

As AI reshapes the global memory infrastructure, the Gulf can be a test bed for the development of responsible and culturally appropriate AI. What emerges here can model how other regions of the world digitize, interpret, and circulate culture in the decades to come. The task now is to build with care, legislate with foresight, and innovate with communities—not just around them.

Finally, as Gulf institutions invest in AI and heritage research, it is critical to avoid the "extraction" of humanistic expertise—a process whereby humanities insights are borrowed for technological advancement without sustained support for the disciplines that produce them. True innovation will depend on a long-term partnership between technical development and humanistic critical reflection.