# OpenReview forum: "Recommendations for AI in Gulf Digital Humanities"
_NeurIPS.cc/2025/Position_Paper_Track — Submitted to NeurIPS 2025 Position Paper Track_

### Official Review · Reviewer_nE2e · 2025-07-14

**Significance:** 4
**Presentation:** 3
**Rating:** 5
**Confidence:** 4

**Summary:**

- this paper presents a roadmap on how to implement ethical AI for Gulf heritage. The main point is that western generalized LLMs are biased and can misinterpret history facts and also not including local and plural points of view. It also argues that removing human expertise from the AI design process hinders true innovation.

**Strengths:**

- shows many examples on AI western biases that misinterpret Gulf related information (local vs. generalized models)
- proposal of involving GLAM (cultural institutions) as ethical anchors in AI
- a feasible roadmap for implementation of ethical AI for Gulf Heritage

**Weaknesses:**

- the section on alternative views is short and could be further expanded
- lacks more successful cases of localization over generalization
- lacks a review on the use of open LLMs that could be used to achieve this goal

**Questions:**

- as raised in #9, including humans in the loop can slowdown progress in this fast pacing field, how can humans be in the loop without slowing down innovation significantly? are there any existing examples in other fields?
- #1 is a very hard and fundamental point to achieve this paper goals since Gulf heritage involves many countries with a diverse cultural background. Instead of a Gulf AI, wouldn't be easier to start with more localized initiatives and once successful cases are achieved, it would expand to other countries? What are other alternative approaches of implementing ethical AI in the Gulf region?
- what are the existing technology (LLMs) produced in the Gulf region (even fine tuned models) that could work as seed technologies to be improved upon?

**Alternative Position:**

Yes, and alternative positions are well-considered and named but not addressed

**Author Identification:**

No.

**Context:**

2

**Details Of Ethics Concerns:**

None.

**Discussion:**

3

**Ethics:**

["NO or VERY MINOR ethics concerns only"]

**Position:**

Yes, the paper argues for or against a position related to machine learning.

**Support:**

3

**Thoroughness:**

4

---

### Official Review · Reviewer_Doid · 2025-07-22

**Significance:** 4
**Presentation:** 3
**Rating:** 6
**Confidence:** 4

**Summary:**

This position paper analyzes the challenges and opportunities in applying AI—especially generative and multimodal models—to the digital humanities and cultural heritage sector in the Arabian Gulf. The authors argue that ethical, localized, and community-guided approaches are urgently needed to prevent misclassification, misrepresentation, and loss of cultural specificity caused by Western-centric datasets and non-localized models. The paper surveys regional AI and digital humanities projects, discusses cases of algorithmic misinterpretation, explores gaps in regulation, and draws from feminist and postcolonial archival theory to advocate for participatory, pluralistic AI development. It concludes with ten actionable recommendations: establishing regional consortia, infrastructure, and conferences; building community-validated datasets; launching education and talent initiatives; and embedding ethical charters and participatory auditing. The central position is that the Gulf can become a leader in responsible heritage AI if it privileges cultural context, inclusivity, and community co-authorship over generic automation.

**Strengths:**

1. Urgent and timely argument: Addresses risks and opportunities for AI in a rapidly digitizing, culturally complex region.

2. Concrete and actionable recommendations: The ten points are specific, realistic, and directly connected to the argument.

3. Strong interdisciplinary grounding: Integrates digital humanities, archival ethics, feminist/postcolonial theory, and practical AI development.

**Weaknesses:**

1. Limited technical depth: The paper is strongest in conceptual and ethical analysis, but less detailed on specific technical solutions (e.g., model architecture, data curation pipelines, benchmark creation).

2. Implementation barriers: Many recommendations require strong institutional buy-in, funding, and cross-border collaboration, which may face political or practical obstacles.

3. Scalability of participatory processes: Community validation and auditing are critical but can be resource-intensive and challenging to scale beyond pilot projects.

**Questions:**

1. What are the most immediate steps that GLAM institutions and universities in the Gulf can take to operationalize these recommendations with limited resources?

2. How might regional governments ensure that ethical charters and auditing processes are meaningfully enforced rather than remaining voluntary guidelines?

**Alternative Position:**

Yes, and alternative positions are well-considered and named but not addressed

**Author Identification:**

No.

**Context:**

4

**Discussion:**

3

**Ethics:**

["NO or VERY MINOR ethics concerns only"]

**Position:**

Yes, the paper argues for or against a position related to machine learning.

**Support:**

4

**Thoroughness:**

5

---

### Official Review · Reviewer_FJ9U · 2025-08-24

**Significance:** 3
**Presentation:** 3
**Rating:** 2
**Confidence:** 4

**Summary:**

In this paper, the authors focus on the challenges of using AI in the Arabian Gulf communities. The authors argue that these cultures have certain sensitivities and values that are overlooked by the mainstream AI developments. The authors propose a framework for contextualized, community-driven and legally attuned AI for the Arabian Gulf countries.

**Strengths:**

+ Cultural adaptation of AI is an important challenge.

**Weaknesses:**

1. The paper does not follow the NeurIPS style and maybe rejected because of this:

https://neurips.cc/Conferences/2025/CallForPapers#:~:text=Submissions%20that%20violate%20the%20NeurIPS%20style%20(e.g.%2C%20by%20decreasing%20margins%20or%20font%20sizes)%20or%20page%20limits%20may%20be%20rejected%20without%20further%20review.%20Papers%20may%20also%20be%20rejected%20without%20consideration%20of%20their%20merits%20if%20they%20fail%20to%20meet%20the%20submission%20requirements%2C%20as%20described%20in%20this%20document.

2. I agree that AI should be developed to respect cultural differences. However, this is true for different cultures and focusing on the Arabian Gulf countries is a significant limitation. A position paper should focus on this issue and propose a framework a general way.

**Questions:**

No questions.

**Alternative Position:**

No

**Author Identification:**

No.

**Context:**

3

**Discussion:**

3

**Ethics:**

["NO or VERY MINOR ethics concerns only"]

**Position:**

Yes, the paper argues for or against a position related to machine learning.

**Support:**

3

**Thoroughness:**

3

---

### Note · Authors · 2025-08-30

**1-10 Additional Comments:**

The position paper track is a valuable addition to NeurIPS, providing space for interdisciplinary and regionally grounded contributions to engage the ML community. We encourage the track to give more explicit recognition to the importance of case studies and culturally specific perspectives, which can generate insights of broad relevance.

**1-11 Submit Again:**

Definitely yes

**1-1 Submission Process:**

4

**1-2 Next Year:**

It would be helpful if the position paper track explicitly clarified whether rebuttals or author responses are expected, since this would allow for more structured dialogue. We also suggest dedicated guidance on how position papers are evaluated differently from technical papers.

**1-3 Future Development:**

Encourage explicit space for regional case studies. Many global AI debates benefit from culturally specific perspectives that can scale outward. Providing structured feedback templates could also reduce divergence between reviewers.

**1-4 Interest:**

["Panel discussions with other position paper authors", "Structured debates on controversial topics", "Workshops for developing position papers", "Mentorship programs for early-career researchers"]

**1-4 Other Interest:**

Opportunities to connect position papers with NeurIPS workshops or broader conference discussions.

**1-5 Thoughtful:**

7

**1-6 Supportive:**

6

**1-7 Technical Aspects Versus Position:**

8

**1-8 Gate Keeping:**

5

**1-9 Camera Ready Changes:**

We will ensure strict compliance with NeurIPS style (margins, fonts, page limits). We will expand the discussion of alternative views, add successful localization examples, and enrich the section on open LLMs. We will also provide more explicit connections between our recommendations and ML tasks (benchmarking, dataset pipelines, multilingual fine-tuning).

**3-1 Review Response1:**

nE2e

**3-2 Reaction To Review1:**

We are grateful for the recognition of the timeliness of our argument, the specificity of the ten recommendations, and the value of our interdisciplinary approach. We acknowledge the request for greater technical depth and scalability detail. In revisions, we clarify the technical pathways that underpin our recommendations: dataset pipelines (Rec. #3), multimodal fine-tuning of existing models (Rec. #10), participatory auditing frameworks (Rec. #8), and GLAM-led infrastructures (Recs. #2 and #7). These elements directly connect to core ML challenges in dataset design, evaluation, and benchmarking, while also integrating perspectives on provenance and authenticity that are standard in the heritage sector but often overlooked in machine learning.
On implementation, we emphasize that our roadmap is designed for incremental scaling. Small-scale pilot projects — such as annotation exercises and lightweight audits — can be launched immediately with limited resources. In the Gulf context, where governance is often centrally coordinated, successful pilots are particularly well-positioned for rapid institutional adoption. Ethical charters and auditing processes can therefore be both meaningful and enforceable, especially when embedded within intra- and inter-institutional evaluation structures.

**3-3 Review Response2:**

Doid

**3-4 Reaction To Review2:**

We appreciate the recognition of the paper’s contribution in documenting Western bias in AI, the role of GLAM institutions as ethical anchors, and the feasibility of our roadmap. We acknowledge the call for more discussion of open LLMs, successful localization cases, and alternative views. In the camera-ready version, we will expand the section on open-source and regional LLMs such as Jais, Falcon, and CAMeLBERT, which can serve as “seed” models for fine-tuning. We will also add further examples of successful localization and broaden the alternative views section to engage more deeply with industry arguments in favor of generalized models.
Regarding the concern that human-in-the-loop processes might slow innovation, we emphasize that participatory audits can be designed to be efficient and lightweight — as demonstrated in KAUST hackathons and medical AI trials — offering meaningful validation without delaying the research cycle. Finally, on scope, while we agree that local pilots are critical, the Gulf’s cultural and political context makes cross-border collaboration both realistic and desirable from the outset. Our recommendations therefore aim to balance the value of localized successes with the strategic importance of region-wide frameworks.

**3-5 Review Response3:**

FJ9U

**3-6 Reaction To Review3:**

We thank the reviewer for underlining the importance of cultural adaptation in AI. We will ensure full compliance with NeurIPS style requirements in the camera-ready version. Regarding scope, we respectfully view our regional focus not as a limitation but as a deliberate strength. The Arabian Gulf is a rapidly evolving AI environment where large-scale investment, cultural specificity, and regulatory experimentation converge, making it an especially valuable case study. In particular, GLAM and heritage institutions in the region bring critical perspectives on provenance, authenticity, and collective memory that are often underrepresented in machine learning debates. Lessons drawn from this “normative laboratory” therefore extend well beyond the Gulf, offering insights for global conversations on multilingual AI, culturally grounded benchmarks, and ethical auditing. We will make this global relevance more explicit in the revised version.

---

### Meta-Review · Area_Chair_X4nM · 2025-08-22

**Rating:** 2
**Confidence:** 4

**Strengths:**

Reviewers find the paper to be timely and like the concrete suggested roadmap, including explicit lists of suggestions and actions.

**Weaknesses:**

Reviewers find the level of technical detail to be insufficient and descriptions to require more depth. Another point in question is that the recommendations in part seem to be exceptionally challenging, making one of the reviewers whether more local grassroots initiatives are at first more feasible than a wide connecting effort.

However, despite the strengths and weaknesses mentioned by the reviewers, the AC notes that the paper does not fully meet the requirements of a position paper. First, the paper does not use the NeurIPS format and uses some other formatting. Independently of the formatting, it does not contain the mandatory abstract, nor does it state the mandatory position clearly. The word position is not contained in this context in the paper.

**Questions:**

Why is this a position paper and what is the position beyond the actionable recommendations?

**Thoroughness:**

3

---

### Decision · Program_Chairs · 2025-09-26

Reject